# Spotting anomalous trades in NFT markets: The case of NBA Topshot

**Konstantinos Pelechrinis**⊙*, **Xin Liu, Prashant Krishnamurthy, Amy Babay**

Department of Informatics and Networked Systems, University of Pittsburgh, Pittsburgh, PA, United States of America

* kpele@pitt.edu

## Abstract

Non-Fungible Token (NFT) markets are one of the fastest growing digital markets today, with the sales during the third quarter of 2021 exceeding $10 billions! Nevertheless, these emerging markets—similar to traditional emerging marketplaces—can be seen as a *great* opportunity for illegal activities (e.g., money laundering, sale of illegal goods etc.). In this study we focus on a specific marketplace, namely NBA TopShot, that facilitates the purchase and (peer-to-peer) trading of sports collectibles. Our objective is to build a framework that is able to label peer-to-peer transactions on the platform as anomalous or not. To achieve our objective we begin by building a model for the profit to be made by selling a specific collectible on the platform. We then use RFCDE—a random forest model for the conditional density of the dependent variable—to model the errors from the profit models. This step allows us to estimate the probability of a transaction being anomalous. We finally label as anomalous any transaction whose aforementioned probability is less than 1%. Given the absence of ground truth for evaluating the model in terms of its classification of transactions, we analyze the trade networks formed from these anomalous transactions and compare it with the full trade network of the platform. Our results indicate that these two networks are statistically different when it comes to network metrics such as, edge density, closure, node centrality and node degree distribution. This network analysis provides additional evidence that these transactions do not follow the same patterns that the rest of the trades on the platform follow. However, we would like to emphasize here that this does not mean that these transactions are also illegal. These transactions will need to be further audited from the appropriate entities to verify whether or not they are illicit.

# 1 Introduction

Non-Fungible Token (NFT) refers to a unit of data that is unique and non-interchangeable. NFTs usually refer to digital items that can be easily reproduced such as, images, text, audio, video etc. Despite the fact that these digital items can be easily copied, their non-fungibility stems from the fact that they are stored on a public ledger (blockchain) and are minted through smart contracts. Every time there is an *event* associated with the NFT (creation, trading etc.) a piece of code stored in the underlying smart contract of the blockchain used to

**Data Availability Statement:** For our study we use a publicly available dataset that can be found at: https://www.kaggle.com/chigorin/nba-topshot-transactions. All the code used to analyze the data and produce the results presented above can be found in the following github repository: https://

github.com/kpelechrinis/NBA-topshot-anomalous-trades.

**Funding:** The author(s) received no specific funding for this work.

**Competing interests:** The authors have declared that no competing interests exist.

manage the NFT is executed. This process enables transparency and tracking of ownership and authenticity of an NFT. It is the equivalent to art forgery detection, only automated and scalable.

NFT marketplaces allow the purchase and sale of these digital tokens/assets. According to Kireyev and Evans' classification [1] there are two types of NFT markets: (i) streamlined and (ii) augmented. Streamlined markets, such as OpenSea and Rarible, are "general-purpose" markets that allow for auctions and fixed-price sales and include a wide spectrum of NFTs. Augmented markets, such as NBA Topshot, are specialized markets that gear more towards value-added services for the participants and focus on a *niche* area. These marketplaces have seen an exponential growth during the last year, with sales during the third quarter of 2021 exceeding $10 billions [2]!

However, as is the case with several traditional marketplaces, emerging markets can be seen as a perfect vehicle for illicit activities such as money laundering. The reliance of NFT marketplaces on the same technology that drives bitcoin and other cryptocurrencies that have been linked in the past with illegal activities on platforms like Silk Road [3–5] makes the case against them even worse. In fact, former SEC officials have expressed their concerns that NBA's NFT platform, Topshot, is ripe for money laundering [6].

In this work, utilizing a large dataset of transactions on the NBA Topshot platform, we design a system for labeling transactions as **anomalous**. Our system includes a sales profit model that uses features of the asset being sold to predict the expected profit to be made for the token. Large deviations from the expected price can serve as a signal for an anomalous transactions. To quantify the latter we model the distribution of the residuals of the profit model using a random forest model for conditional density estimation (RFCDE) [7]. This allows us to estimate the probability of observing a profit as high as the one observed in the data, and label transactions whose corresponding estimated probability is less than 1% (or in general any predefined threshold) as anomalous.

Of course, we do not have the ground truth on whether each transaction is anomalous or not in order to perform traditional evaluations. However, we use the transactions labeled by our framework as anomalous to create a directed trade network of who-trades-to-whom and compare its structure with the full trade network. The results indicate that these two networks are very different in terms of the network metrics examined, namely, edge density, transitivity, degree distribution and node centrality. While this again does not serve as a hard evaluation of our anomalous labels, it provides additional evidence that strengthen our belief that these transactions are indeed *different* and unusually profitable when compared to the rest of the transactions on the platform. We would like to make it clear that given the type of information publicly available (and described in the following sections) we cannot make any claim beyond a transaction being much different than *normal*, and hence, our system should be treated as a first line of defense, that is, flagging transactions for further financial inspection.

The research on these markets has not followed suit yet and has mainly focused on the underlying blockchain protocols and standards, copyright regulations and their impact on the world of art [8–12]. Empirical analysis of the properties and structure of NFT markets is still limited and has mainly focused on specific platforms and the relationship between pricing, scarcity and cryptocurrencies [13–15]. Recently, Nadini *et al.* [16] provided the first comprehensive overview of the NFT market by analyzing the statistical properties of a large scale NFT market that includes several different categories of tokens. They also analyzed the trade networks and found that traders typically are *specialized*, that is, focused on specific types of tokens and form tight clusters with other traders that trade similar tokens. They also develop a linear regression model for the price of future primary and secondary sales of NFTs. Kapoor *et al.* [17] further investigate the relationship between social media (in particular Twitter)

promotion of an NFT and its sales price and they find that social media features improve the prediction of the valuation. On a tangential direction, Franceschet [18] use the network between sellers (artists) and buyers (collectors) to rate artists and collectors based on network centrality metrics, while also providing investment strategies with different risk/reward profiles based on a variety of network metrics.

## 1.1 Anomaly detection in transactions, markets and networks

There is a huge line of research that focuses on identifying anomalies in data representing a wide range of relationships, from transactions (financial or otherwise) to interactions between people and/or organizations. Network analysis has been used to spot outliers in settings including fraudulent reviews on online platforms, outlier donations to political parties and candidates, abuse in healthcare claims and medical prescriptions [19–22]. Machine learning (supervised or unsupervised) has also been used to identify financial crimes, money laundering and in general financial transactions that are *suspicious*. Credit card fraud is one of the most common financial crimes and financial institutions are trying to find accurate ways to identify suspicious transactions. As a result a large volume of research for detecting credit card fraud using machine learning and artificial intelligence methods exists ([23, 24] provide comprehensive surveys on this line of research, while Stojanović *et al.* [25] provide a comprehensive survey on fraud detection in general fintech applications).

While at a high level our framework borrows ideas from the existing literature, we put more emphasis on the uncertainty associated with both (i) the underlying process we are trying to capture (i.e., profit from sales), as well as, (ii) the models developed for this prediction task. In particular, we believe that the use of RFCDE (described in Section 2.2.2) for estimating the probability density of the profit can open new directions and applications that go beyond that traditional point estimate models. Finally, we combat the absence of ground truth for each transaction by obtaining additional, corroborating, evidence for the status of the identified anomalous transactions through a different approach, namely, network analysis of the trade network.

In brief the contributions of our study are as follows:

- Provide the first—to our knowledge—comprehensive analysis of the NBA's TopShot NFT marketplace

- Provide a novel synthesis of modeling approaches that allows us to essentially estimate the probability distribution of the profit from a collectible's sale, and hence, the probability of a transaction leading to the observed profit.

- In the absence of ground truth for the state of transactions, we provide robustness checks based on appropriate analysis of the underlying trading networks, which corroborates our labeling of transactions as anomalous.

The rest of the paper is organized as follows: Section 2 describes in detail the data we used and our modeling framework, while Section 3 provides the results of our analysis along with the network analysis of trade networks. Finally, Section 4 concludes our work by discussing as well its limitations.

## 2 Materials and methods

### 2.1 Data description and exploration

Topshot is an NFT market that runs on top of the Flow blockchain. The dataset we used for our study includes information from transactions that took place on the platform between 07/27/2020 (the first day of operation of topshot) and 03/19/2021 (see Table 1 for some basic

**Table 1. Basic statistics for the Topshot market.**

| # of users | 245,099 |
|---|---|
| # of transactions | 2,631,731 |
| Avg. value per transaction ($) | 152.8 |
| Median value per transaction ($) | 28 |

statistics of these transactions). This covers the majority of the most "active" period on the platform to date as shown in Fig 1, which shows the total lifetime daily sales on the platform (as of the time of writing). As we can see the period between the end of January 2021 and early April 2021 exhibits the highest activity volume on the platform in terms of monetary sales. In particular, during that period that covers only 13.5% of topshot's lifetime the platform realized 57.5% of the total lifetime sales volume. While the lower volume of sales at the beginning of the platform's operation was a result of the private, invitation-only, initial launch [26], the slowdown after April 2021 could be attributed to a variety of reasons, ranging from scalability issues [27] to the inability of users to withdraw the money from their accounts [28].

Each transaction data point is a tuple with the following information/format: `<moment_unique_id, moment_id, player_id, set_id, seller_id, buyer_id, play_category, limited_flag, circulation_count, transaction_time, transaction_id, sale_price>`. Every clip/collectible is associated with a `moment_id`, while each copy of the clip has its own `moment_unique_id`. The latter allows us to track the transactions involving a specific collectible, and hence, calculate the amount of times it has been traded in the past, as well as, the prices for which it has been sold/ traded. The `limited_flag` feature is a binary attribute that specifies whether the moment sold was in limited quantities, while the `circulation_count` further specifies the number of copies for the token.

**2.1.1 Exploratory analysis.** Before delving into the details of identifying abnormal transactions, we explore the data to get a better understanding of the basic structure of the market.

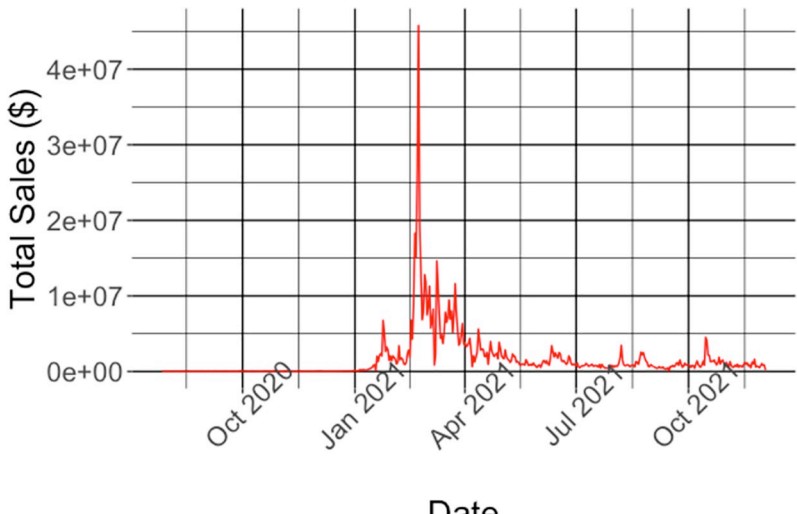

**Fig 1. The majority of the activity on TopShot, in terms of total monetary value, happened during February-March 2021.**

We begin by looking at the distribution of transactions performed per user during the period covered from our data. We examine separately sales and purchases. Fig 2 presents the results, where as we can see the distribution for both sales and purchases per user exhibit a heavy tail. The tail of these distributions usually are described through and conform to a power law, i.e., the probability $p_k$ of observing $k$ sales (or purchases) is given by $p_k \propto \dfrac{1}{k^{\alpha}}$, for $k > k_{min}$. However, in our setting a double power law [29] seems more appropriate to describe the distributions at hand. A double power law essentially exhibits multiple regimes in its tail, each of which with different scaling (i.e., exponent). As we can see, in our case there are two regimes in the tail of the distribution, separating two different power laws. In both sales and purchases distribution the end tail of the distribution exhibits a faster drop as compared to its earlier part.

While long tail distributions have been central to describing complex systems, these typically include a single scale. However, double power laws have also been observed in various settings [30], but the process underlying their emergence has not been studied extensively. Almost all of this line of work focuses on degree distributions in networks, where node aging effects [31] or a combination of linear preferential attachment with a uniformly random process of closing triangles in the network [29] can lead to two distinct scales at the tail. In the case of TopShot sales/purchases, the two separate scales can be a result of the different evolution of the platform during different time periods. In fact, the plausibility of this hypothesis is further strengthened by the single exponent identified in both distributions when analyzing only the data prior February 2021. The hypothesis is that early adopters and the users that joined the platformed after its meteoric growth exhibit different behavior. Nevertheless, identifying the actual mechanism driving these distributions is beyond the scope of this paper.

However, central to our analysis are the prices each collectible is traded for, as well as, the profits users make off of them. A user can acquire collectibles either directly through the system by buying what it is called "packs", or through the peer-to-peer market available on the

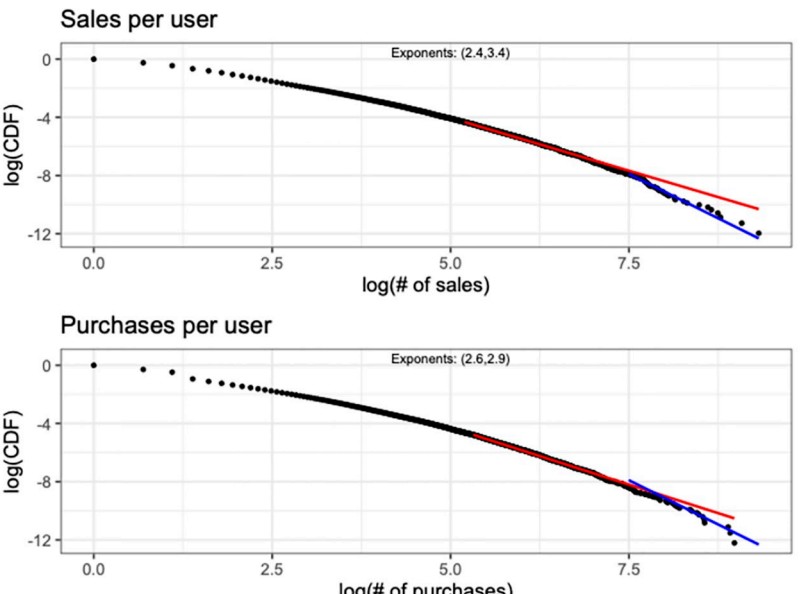

**Fig 2. The number of sales and purchases per user can be both described from a double power law.**

platform from other users. Our data only include the transactions on the peer-to-peer market and hence, we cannot calculate the total net profit a user had during the period our dataset covers, since we do not know how much—if any—money the user spent on packs. However, we can calculate the profits users made through trading collectibles they acquired from other users. Given that each collectible has a unique moment id, we can trace its ownership chain and hence, calculate the profit a user made by buying and then selling it. We will term this as the *flip profit*. The distribution of the flip profit exhibits a (very) long tail, with a very small fraction of users making an extraordinarily large flip profit. However, overall the median flip profit is $31, while the average is approximately $1,420 (again due to the long right tail). In particular, there are 103 users (or 0.16% of the users) that have made a total flip profit of more than $100K during the period covered and 278 users (or 0.44% of the users) that have made a flip profit of more than $50K. Furthermore, there are only 7,608 users (or approximately 12% of the users) that have made a profit larger than the average flip profit. Finally, about 22.6% of the users have a total loss from their collectible flips, with an average loss of of approximately $200 (median loss $39).

## 2.2 Profit model and profits above expectation

Central to our approach is a model $\mathcal{M}_{PE}$ that estimates the profit $\hat{p}_i$ that the sale of a collectible is *expected* to provide based on a variety of attributes. This will allow us to consequently estimate the profit above expectation $\texttt{PAE}_i$ that the seller made as $\texttt{PAE}_i = p_i - \hat{p}_i$, where $p_i$ is the actual profit made from the sale of collectible $i$. $\mathcal{M}_{PE}$ can provide us with a point estimate (thus, *PE*) for $p_i$—and consequently for $\texttt{PAE}_i$—and an initial insight with regards to whether a specific transaction is anomalous/suspicious or not. In our study, we use a linear regression for $\mathcal{M}_{PE}$. The main reason for this choice is its interpretability.

However, simply the fact that a sale provided a large profit, it does not automatically justifies labeling it as suspicious or even anomalous. After all $\hat{p}_i$ is just the expected value of the profit from the sale of collectible $i$ and it is not necessarily the most probable one. There is an error associated with the point estimate prediction that allows for the possibility of the true profit being (much) higher or lower. Therefore, a more robust way for labeling a transaction as anomalous is to consider this uncertainty. We achieve this by building a second model, $\mathcal{M}_{CDEres}$, for the **distribution** of the residuals of $\mathcal{M}_{PE}$. This allows us to estimate the uncertainty of the predicted profit $\hat{p}_i$, and consequently the probability of a profit (or loss) at least as high as the one observed in real data. For $\mathcal{M}_{CDEres}$ we use RFCDE [7], a random forest model for estimating the conditional density estimation of a response.

**2.2.1 Expected profit model $\mathcal{M}_{PE}$.**   We start by describing the linear regression model for the expected profit from a sale. The response variable is simply the difference between the sale price of the collectible and the price it was previously bought by the seller. The independent variables of our model are described below:

**Circulation Count**: This is the number of copies available on the platform for the specific collectible at the time of the sale.

**Limited Edition**: This is a binary variable that specifies whether the collectible sold was designated as limited edition (LE) during the sale. For collectibles that are designated as LE there will be no more copies of it minted from the platform, and hence, their value is expected to increase.

**Serial Number**: This is the serial number of the collectible. The serial number specifies the sequential order in which each parent collectible was minted in. For example, if the serial number of a specific LeBron James collectible with a circulation count of 15,000 is 132, it means that it was the 132th collectible created on the block chain. In general, collectibles with lower

serial numbers are associated with higher prices [32] and hence, it is only natural to make the hypothesis that the same goes for profits.

**Play Category**: This is a categorical variable that identifies the type of play included in the collectible. It can be any of the following types: Assist, Block, Dunk, Handles, Jump Shot, Layup, Steal or 3 Pointer.

**Player**: This categorical variable captures player's effects on the profits from the resale of a collectible. There is a total of 296 players for which there are topshot collectibles in our dataset.

**Trade counts**: This represents the number of times this specific collectible has been traded in the past.

**Bought Price**: This is the price the seller had bought the collectible for.

**Comparable Profit**: This is the average profit made by the chronologically last 10 sales from a copy of the same collectible.

We also include in the regression two interaction terms; one between the circulation count and the limited edition identifier and one between the bought price and the comparable profit.

Table 2 shows the regression coefficients (for readability we have excluded the regression coefficients for the player effects, since there are 295 of them as explained earlier). As we see this simple linear model explains approximately 73% of the variance in the profits observed in our dataset, while the standard error is approximately $308. Fig 3 shows the residuals plot for

**Table 2. Regression table for $\mathcal{M}_{PE}$.** Coefficients for player effects are not shown for readability.

|  | Dependent variable: |
|---|---|
|  | **Profit** |
| Circulation Count | 0.006*** (0.0005) |
| Limited Edition | 124.488*** (16.791) |
| Serial Number | −0.002*** (0.0001) |
| Play_category:Assist | 9.864*** (1.986) |
| Play_category:Block | 1.096 (2.760) |
| Play_category:Dunk | −4.850*** (1.665) |
| Play_category:Handles | 12.396*** (2.484) |
| Play_category:Jump Shot | −9.913*** (2.020) |
| Play_category:Layup | −8.751*** (1.828) |
| Play_category:Steal | 0.102 (3.653) |
| Trade Count | −11.214*** (0.348) |
| Comparable Profits | 1.108*** (0.001) |
| Bought Price | 0.174*** (0.001) |
| Circ Count × Limited Ed | −0.001** (0.0005) |
| Comp Profits × Bought Price | −0.00002*** (0.00000) |
| Intercept | −157.002*** (18.107) |
| Observations | 1,025,728 |
| $R^2$ | 0.727 |
| Adjusted $R^2$ | 0.727 |
| Residual Std. Error | 308.379 (df = 1025420) |
| F Statistic | 8,878.443*** (df = 307; 1025420) |

*Note*:

*p<0.1;

**p<0.05;

***p<0.01

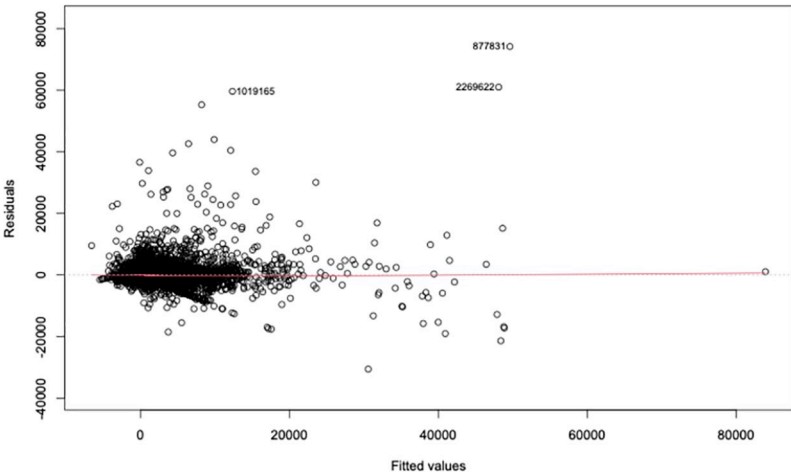

**Fig 3. The residuals are clustered around 0, but the distribution appears to exhibit skew and/or heteroscedasticity.**

the model. As we can see the residuals are clustered around 0, however the distribution appears to have some skew or even exhibit heteroscedasticity. This essentially means that while the regression coefficients are still unbiased the estimation of the standard errors can be in error. This does not impact the point estimates of the dependent variable, which to reiterate are still unbiased, but it can impact the estimation of its prediction interval, which assumes that the errors follow a normal distribution with 0 mean and constant variance. The prediction interval is particularly important for labeling a transaction as anomalous since it allows us to estimate the probability of observing a profit (or loss) as high as the one in the data. In what follows we describe $\mathcal{M}_{CDEres}$, an random forest model that allows us to estimate the empirical distribution of the residuals and hence, obtain prediction intervals for the profit from a transaction.

**2.2.2 Model $\mathcal{M}_{CDEres}$.** In order to model the residuals of the $\mathcal{M}_{PE}$ we will use RFCDE [7]. RFCDE has applications in settings with nonstandard error distributions and multimodal or heteroskedastic response variables, offering a more subtle way to quantify uncertainty in these situations. With RFCDE we can approximate the probability distribution function of the residuals, allowing us to make probabilistic inferences for the residuals of $\mathcal{M}_{PE}$—and consequently the profit. RFCDE uses the CDE loss [33] to partition the feature space and then construct a weighted KDE estimate of the response with weights defined by leaves in the random forest. In our case, we will use a single variable for the conditional density estimation, namely, the predicted value $\hat{p}$ for the profit from $\mathcal{M}_{PE}$.

With $\hat{f}(r|\hat{p})$ being the conditional density estimate of the residual $r$ we can now estimate the probability of a sale making at least a profit of $\pi > 0$ (similarly for a loss) as:

$$\Pr[p > \pi] = \Pr[r + \hat{p} > \pi] = \Pr[r > \pi - \hat{p}] = \int_{\pi - \hat{p}}^{\infty} \hat{f}(r|\hat{p})\, dr \qquad (1)$$

We consequently use this probability to label a transaction as anomalous or not. If the probability of a transaction making at least the observed profit is *extremely low*, then we can label this transaction for further auditing (from the appropriate entities). However, what probability is considered extremely low? The choice of this threshold has a direct consequence on how conservative the framework is at flagging transactions as anomalous. It is very similar to the choice of significance level of a hypothesis test. While there is not a formal way to those the

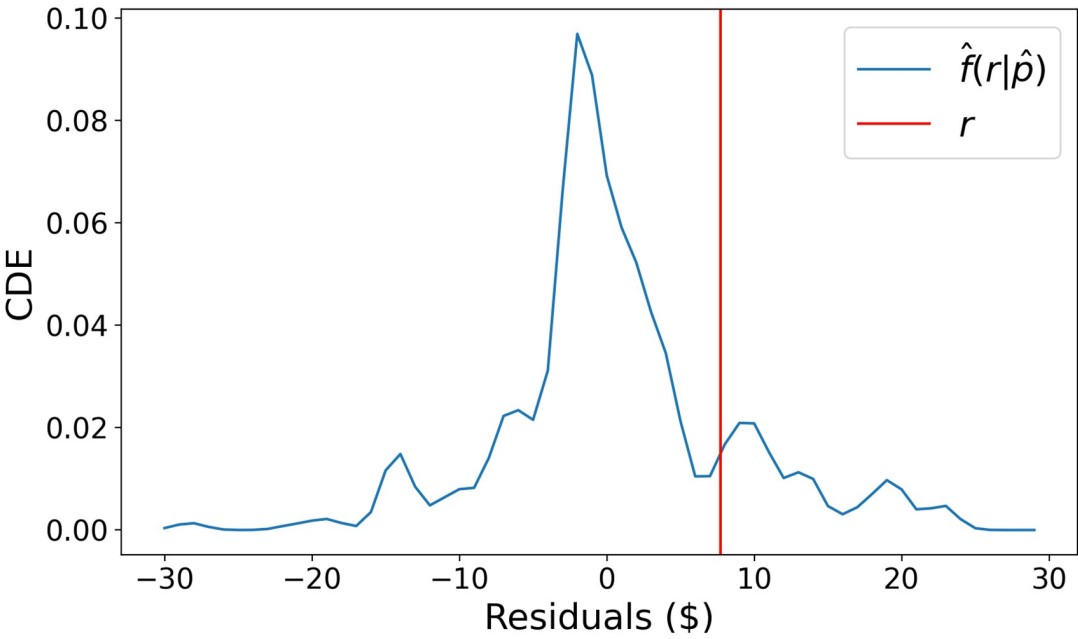

**Fig 4. Estimated conditional density for the residual of the expected profit from the sale of the Dennis Schroder collectible.**

threshold for these tests, we typically use a significance level of 5% (or 1% or 10%). These values, despite being used in almost all of scientific research, they are arbitrarily chosen. The most appropriate approach is to provide the actual probability value (without a threshold to quantize the decision) and allow the end user of the downstream application decide whether this probability is high or low. For illustration purposes for this study we will use a threshold of 1% probability, but to reiterate, our recommendation is to not consider a "one-size fits all" threshold but consider the specific application/data at hand.

Let us see an example, in order to better understand the labeling process. A Dennis Schroder layup moment that was bought from user `Diogolos` for $24, is expected to be sold for an expected profit of $2.3. It is eventually sold to user `Baron_Von_Levetron` for $31, and a profit of $7. The question we are trying now to answer is what is the probability that this collectible would be sold for $31 or more, or that the residual of $\mathcal{M}_{PE}$ would be larger than $7-$2.3 = $4.7. Fig 4 shows the estimated CDE for the residual for this prediction, while with the vertical line the *true* residual is marked. Using this CDE we get a probability of observing a profit above expectation of at least $4.7 as 30%, and hence, this transaction is not labeled as anomalous.

In the following section we present the results of our analysis. In the absence of ground truth, and hence, *hard* evaluation metrics, we will take an indirect way in evaluating our analysis. In particular we will use network science metrics to identify any differences in the trade networks involving the transactions labeled as anomalous and those that were not. Network analysis has been shown to be able to distinguish between illicit and non-illicit activities in an organization [34].

## 3 Results

We begin this section by reporting the results of our transaction labeling. Out of the 1,025,728 peer-to-peer trades in our dataset, 2,767 of them exhibited a positive PAE and were labeled as

**Table 3. Sample of transactions labeled as anomalous.**

| Player | Bought for ($) | Sold for ($) | PAE | $p_{PAE}$ |
|---|---|---|---|---|
| P. Achiuwa | 1,005 | 11,999 | 10,286.88 | 0.009 |
| M. Smart | 33 | 800 | 696.72 | 0.009 |
| J. Green | 3 | 573 | 488.69 | 0.008 |
| L. James | 1,199 | 125,000 | 74,184.40 | 0.005 |
| B. Adebayo | 5,999 | 49,999 | 39,675.72 | 0.008 |
| J. Embiid | 15 | 99 | 69.5 | 0.007 |

anomalous from the framework presented in the previous section. It is worth noting that while our dataset includes more transactions, we cannot track the profit on all of these, since we cannot track the first trade for all the moments in the dataset. Table 3 presents a sample of the transactions that were identified as anomalous. For example, a TopShot moment featuring Bam Adebayo that was bought for $5,999 was sold for $49,999 for a profit above expectation of more than $39,000. The probability of this particular moment giving that much profit above expectation was deemed as 0.8% by our framework, and hence, it was labeled as anomalous.

These results do not mean that the transactions that were labeled as anomalous are all illicit. However, they do not fit the *typical* transaction according to our models and need to be examined further. To this end we build a trade network $\mathcal{N}_\tau$, where the set of nodes $V$ are the users of the platform and there is a directed edge $e = (u, v)$ from user $u$ to user $v$ if $u$ sold a collectible to $v$. We will then compare properties of the full trade network $\mathcal{N}_\tau$, with those of a subnetwork $\mathcal{N}_{\tau,s}$ obtained by considering only the users that have participated in the transactions deemed by our framework as anomalous. As alluded to above, fraudulent or corruption networks exhibit, context-dependent, differences in structure and properties as compared to the corresponding "law-abiding" ones [34–36]. We would like to emphasize here that this approach is not the only one possible. However, it is a rather straightforward approach with a clear intuition; **if a set of transactions that are labeled as anomalous are better connected as compared to the rest of the trade network then our belief on these transactions being anomalous is stronger**, since this can be a good signal for the presence of collusion and increase our confidence in the status of these transactions.

Going back to the sample results presented in Table 3 we can see that not all trades labeled as anomalous are sold for such high prices. For instance, the last row shows a Joel Embiid moment that was bought for $15 and sold for $99 for a PAE of $69.5. While similar transactions have a low probability of providing the PAE they did, it is hard to imagine that they are carriers of illicit activity, at least when considered in isolation. For this we will examine different trade subnetworks based on the value of the anomalous-labeled transactions. In particular, subnetwork $\mathcal{N}_{\tau,s,\delta}$ is obtained by considering only the users that have participated in transactions that were deemed anomalous and were worth at least $\$\delta$. In what follows we will use $\delta \in$ {1, 500, 1000}. Table 4 provides some basic information about the networks we will analyze in what follows. Fig 5 also visualizes the full trade network, $\mathcal{N}_{\tau,s,1}$ and two random subnetwork samples from $\mathcal{N}_\tau$ of similar size to $\mathcal{N}_{\tau,s,1}$. The one of them was chosen to have the same number of edges as $\mathcal{N}_{\tau,s,1}$, while the other was chosen to have the same number of nodes as $\mathcal{N}_{\tau,s,1}$. As we can visually see, and will be verified by various network metrics examined in what follows, the subnetwork obtained from the anomalous transactions exhibits much higher levels of connectivity as compared to the sparser full trade network. This becomes even more clear when we zoom in the random subgraphs of the network of comparable size to $\mathcal{N}_{\tau,s,1}$. We will focus on metrics that quantify various aspects of the network's connectivity.

**Table 4. Basic statistics for the subnetwork induced from the transactions labeled as anomalous for different values of $\delta$.**

| Network | $\mathcal{N}_\tau$ | $\mathcal{N}_{\tau,s,1}$ | $\mathcal{N}_{\tau,s,500}$ | $\mathcal{N}_{\tau,s,1000}$ |
|---|---|---|---|---|
| # of nodes | 159,598 | 3,523 | 1,535 | 799 |
| # of edges | 978,673 | 72,471 | 24,663 | 10,484 |
| Avg. degree | 6.13 | 20.57 | 16.1 | 13.1 |

**Fig 5. The subnetwork $\mathcal{N}_{\tau,s,1}$ induced by the transactions deemed as anomalous has a very different structure compared to the full trade network $\mathcal{N}_\tau$, which is particularly visible when zooming to two random samples of the $\mathcal{N}_\tau$ of comparable size to $\mathcal{N}_{\tau,s,1}$.**

## 3.1 Edge density

We first focus on the edge density $D$ of the network, that is, the fraction of existing edges over all the possible edges in the network, since this will give us an intuition about how tightly connected the different networks are. The original network is extremely sparse with a density of $D(\mathcal{N}_\tau) = 3.8 \cdot 10^{-5}$. The subnetworks obtained from the subset of transactions that were deemed as anomalous are still relatively sparse but they are orders of magnitude denser. In particular, $D(\mathcal{N}_{\tau,s,1}) = 0.006$, $D(\mathcal{N}_{\tau,s,500}) = 0.011$ and $D(\mathcal{N}_{\tau,s,1000}) = 0.016$. To some extent these results were expected given the much higher average degree of the subnetworks extracted by the anomalous transactions as compared to the original trade network. To further assess the robustness of this difference in edge density for the different values of $\delta$ we generate 20,000 random subnets with the same number of nodes as $\mathcal{N}_{\tau,s,\delta}$ and compute their density. We can then calculate the probability of the density at a random subnetwork with the same number of nodes have edge density at least $D(\mathcal{N}_{\tau,s,\delta})$. In our simulations, there was no sampled subnet for all values of $\delta$ that exhibited an edge density as high as the corresponding $\mathcal{N}_{\tau,s,\delta}$. More specifically the average edge density values we obtained for the randomly sampled subnetworks are all approximately $9.2 \cdot 10^{-5}$, which is 2.5 times higher than the edge density of the whole network, but still orders of magnitude smaller compared to the subnetworks from the anomalous-labeled transactions. Simply put the users participating in what our models believe are transactions out of the ordinary form a much denser network as compared to both the full

TopShot trade network as well as random subnetworks of similar size to $\mathcal{N}_{\tau,s,\delta}$. Furthermore, the density of these anomalous transactions subnetworks increases as we increase the value of $\$\delta$. This means that for anomalous transactions of higher value, the network is even more densely connected. While this does not necessarily validate their label as anomalous, it certainly provides additional evidence that the users engaging in these transactions are connected more tightly as compared to the rest of the users/transactions.

## 3.2 Node degree

The degree of a node represents the number of connections the node has in the network. In the case of a directed network, where the relationship is directional (as is the case in our study), there are two different types of degree, namely the in and out degree of the node, depending on the direction of the relationship. The node degree is one of the simplest, yet illustrative, node *importance* as it captures how many relationships a node in the network has. As we see from Table 4 the average degree (recall that the average in and out degree of a directed network are equal) for the anomalous trade networks is much higher compared to the full trade network. In what follows we examine the in, out and total degree distribution of the various networks as well as the max degree for each network (expressed as a fraction of the network size). In all cases, the distribution has a heavy tail that can be approximated by a power-law. The exponents of these power-laws though are different for the full network and the anomalous subnetworks, particularly for the out and total degrees (Table 5).

However, given the vast difference in the network size and the absolute values for the degrees possible, we compare the power law exponent of $\mathcal{N}_{\tau,s,\delta}$ with the distribution for the exponent obtained from a set of 20,000 randomly sampled subnetworks from $\mathcal{N}_\tau$ of the same size. As we can see in Fig 6 in all cases the anomalous networks exhibit on average a larger exponent (marked by the vertical line) as compared to the ones observed from the subnetworks. The empirical probability of the exponents of the random subnetworks being at least as large as the one for the corresponding $\mathcal{N}_{\tau,s,\delta}$ is less than 0.001 in all cases, and hence, we can conclude that they are statistically different. Furthermore the average maximum (in/out) degree observed in the sampled networks—as well as the full trade network $\mathcal{N}_\tau$—is less than 2% of the size of the network, while for the anomalous networks this ranges from 11% to 23%.

In summary, users in the anomalous trade subnetwork are connected to a larger fraction of the network as compared to random subnetworks of the same size. This maximum degree is also a non-significant fraction of the network (larger than 11% in all cases).

## 3.3 Network transitivity

Next we turn our attention to network transitivity. Network transitivity refers to a phenomenon observed in several social networks, where the probability of a pair of nodes being connected increases with the presence of a common connection. Essentially, if nodes A and B are connected, and so are nodes A and C, then, in a network with transitivity, nodes B and C are

**Table 5. The power-law exponent for the degree distributions of the various "anomalous" subnetworks and the full trade network.**

| Network | $\mathcal{N}_\tau$ | $\mathcal{N}_{\tau,s,1}$ | $\mathcal{N}_{\tau,s,500}$ | $\mathcal{N}_{\tau,s,1000}$ |
|---------|--------------------|--------------------------|----------------------------|------------------------------|
| In | 3.3 | 3.7 | 3.5 | 3.6 |
| Out | 2.9 | 3.5 | 3.4 | 3.4 |
| Total | 1.7 | 3.6 | 3.2 | 3.8 |

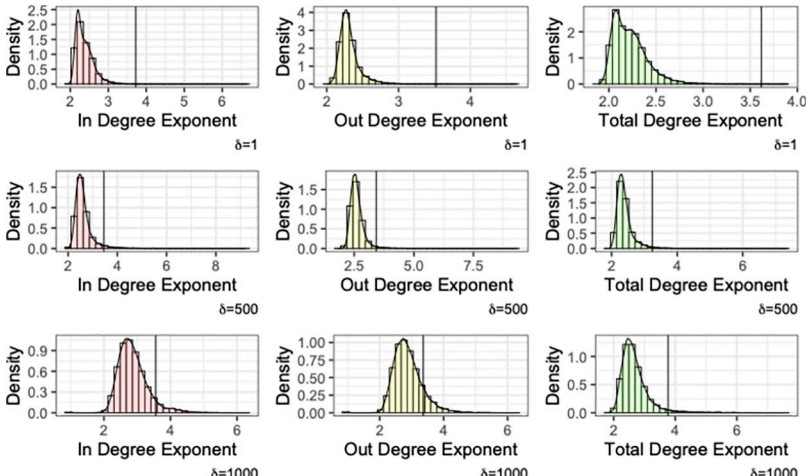

**Fig 6. Bootstrapped distribution for the power-law exponent of the degree distributions for trade networks of the same size as** $\mathcal{N}_{\tau,s,\delta}$.

more probable to be connected as compared to the case where either of the edges A-B or A-C where absent. This results in more triangles forming in the network as compared to the ones expected at random. The (global) clustering coefficient [37] quantifies the prevalence of these triangles by calculating the fraction of closed triples over all triples in the network. This is a measure on how tightly connected the nodes in the network are. While edge density captures the overall density of connections in the network, the clustering coefficient is particularly focused on a local network motif (triangles) that is indicative of tightly knit groups with high density edges in their neighborhood. The clustering coefficient for the full trade network $\mathcal{N}_\tau$ is 0.02; i.e., only 2% of the network triplets are closed. For the anomalous transactions trade networks the clustering coefficients are 0.11, 0.15 and 0.20 respectively for $\delta = \{1, 500, 1000\}$.

Similar to the other network metrics we again sample 20,000 randomized subnetworks from the full trade network of the same size as $\mathcal{N}_{\tau,s,\delta}$. The average clustering coefficient over the different samples are 0.02, 0.02 and 0.01 respectively, while the empirical probability of observing a clustering coefficient in the randomized subnetworks as high as the corresponding $\mathcal{N}_{\tau,s,\delta}$ is less than 0.005 for all values of $\delta$ examined. This means that the full transaction network differs from the anomalous-labeled transactions trade subnetworks in terms of transitivity as well.

## 3.4 Network centrality

We now turn our attention to examining the distribution of node (user) *importance* in the network. We have already examined the degree of a node as a measure of importance earlier, but the node degree assumes that all connections in the network are equally important, thus, boiling down to simply a count of connections. However, this is certainly not the case in most real networks. Trading an NFT to a trader who is *important* themselves, should bear more weight than trading to a newcomer for example. This leads to an iterative definition of importance/ centrality and there are various network metrics that provide us with the node importance based on this high level thought process. In our case, since we have a directed network and users have different "roles" depending on whether an edge emanates from or points to them we will use the HITS algorithm [38] to obtain two separate centrality values that will correspond to buyers and sellers in the trade network. In brief, the centrality values obtained from

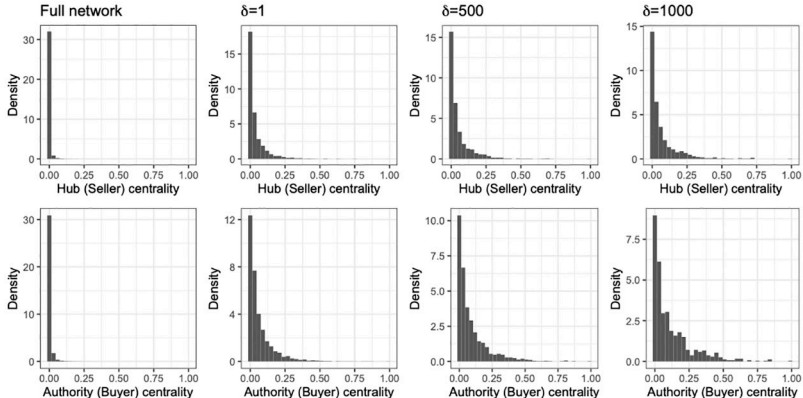

**Fig 7. The distribution of the hub and authority centrality values of the induced subnetworks $\mathcal{N}_{\tau,s,\delta}$ exhibit heavier tails as compared to those in the full trade network.**

the HITS algorithm (namely, hub centrality and authority centrality) are computed in a mutually recursive way. Specifically, the authority centrality of a node $u$ is equal to the sum of the hub centrality values of the nodes pointing to $u$, while the hub centrality of $u$ is the sum of the authority centrality values of the nodes $u$ points to. In our setting, the hub centrality corresponds to a *central* seller in the network, while the authority centrality corresponds to a central buyer. The distribution of centrality values can provide us with insight on whether there are individuals that are really separated from the rest of the network when it comes to their *importance*. Typically centrality measures such as eigenvector, betweeness, and PageRank exhibit a right-skewed degree distribution (not necessarily power-law), while others like closeness centrality are more concentrated around a small range of values [39]. Fig 7 depicts the distribution of the hub (top row) and authority (bottom row) centrality values for the full trade network (first column) and the subnetworks $\mathcal{N}_{\tau,s,\delta}$. As we can see the distribution for both centrality scores for the subnetworks consisting of the transactions labeled as anomalous exhibit much fatter tails. This means that in these networks it is more probable to have nodes that will emerge as central buyers or sellers as compared to the full TopShot trade network.

This can be an artifact of the different network sizes, and hence, similar to the previous network metrics, we examine the robustness of these differences by sampling random subnetworks of the same size as $\mathcal{N}_{\tau,s,\delta}$ and computing HITS. We then perform a Kolmogorov-Smirnov test to compare the distribution of the centrality values in the random subnetworks and the corresponding $\mathcal{N}_{\tau,s,\delta}$. In all of our sampled subnetworks, and for all values of $\delta$, the p-value of the Kolmogorov-Smirnov test is close to zero ($< 0.001$), and, hence, we can conclude that the probability of getting these skewed distributions for the HITS centralities in the anomalous subnetworks purely by chance is extremely small.

To summarize our results from the network analysis of the full trade network, as well as, the subnetworks based on the anomalous-labeled transactions provide additional evidence that these transactions do not follow the *normal* patterns of the rest of the trade network. In particular, these anomalous-labeled subnetworks exhibit much higher edge density and transitivity, as well as, heavier tails for the degree and HITS centrality distributions.

## 4 Discussion and conclusion

In this paper we focus on anomalous trades on a specific NFT platform, namely, NBA's Top-Shot. We start by building a linear regression model for the profit to be made from the sale of a

specific collectible. We consequently model the conditional density of this model's errors using RFCDE, which allows us to estimate the probability that the sale of a specific collectible would generate profit at least as high as the one realized in the data. This allows us to label transactions as anomalous if this probability is extremely low (we use a threshold of 1% in our analysis). Furthermore, in the absence of ground truth for these transactions we compare properties of the full trade network to those of the corresponding subnetworks that include only the transactions labeled as anomalous. The results clearly show that these transactions are not only anomalous in terms of their expected profit based on our model, but are also very different in terms of the underlying trade network structure. Of course, to reiterate, this does not necessarily mean that the transactions are illicit. Similar to most fraud analytics systems, our framework can only provide triggers and flag transactions as potentially illicit/fraudulent that need to be further investigated. This investigation might entail the development and use of a different set of models that utilize more advanced features that might be hard (or costly) to obtain for all transactions (e.g., on a proof-of-stake blockchain extract information about the consensus of each fork). Nevertheless, this step does not have to necessarily be automated but it can also be manual, similar for example to IRS' audits for flagged tax returns.

An alternative approach for estimating the expected profit from a trade would be to directly model the conditional density of the profit using RFCDE. The reason that we did not take this approach is mainly interpretability. In particular, the linear regression model allows us to gain a quantitative understanding of the relationship of each of the collectible's features on the expected profit. Of course our work does not come without limitations. The biggest one is the absence of ground truth that will allow us to perform *hard* evaluation of our approach. However, this is the case with most of the studies that deal with anomaly detection. It is very rare the case that a dataset annotated with true anomalies is available. However, despite this limitation—shared by most of the related literature—we believe that our work provides a sound framework, combining data and network science methods, that can help narrow down the set of anomalous transactions.

One thing that we would like to point out is that our dataset includes all transactions up to 03/19/2021. While the Top Shot market is still active, the activity has declined drastically since 05/2021. In fact, the year-to-year decline in transaction volume has been at 94% [40]. Therefore, our dataset includes the most active period in the marketplace [41], which is also the early adoption phase that we are mainly interested in. During the early adoption phase there are many different types of users that come to the platform to explore it and to eventually become long term users or abandon the platform. Therefore, by analyzing this phase of an innovation one is able to capture all the different types of uses people might have for the innovation (including possible use for illegal activities). Furthermore, since the innovation is still new during this phase, it is also more probable to have "anomalies" that might not necessarily be illicit, but a result of users not having a norm reference for using the platform. This of course does not mean that late adopters cannot find new (or anomalous) uses for an innovation, but in the case of TopShot the user base has shrunk and abandoned the platform altogether.

## Author Contributions

**Conceptualization:** Konstantinos Pelechrinis, Prashant Krishnamurthy, Amy Babay.

**Data curation:** Konstantinos Pelechrinis, Xin Liu.

**Formal analysis:** Konstantinos Pelechrinis.

**Investigation:** Konstantinos Pelechrinis, Amy Babay.

**Methodology:** Konstantinos Pelechrinis, Xin Liu, Prashant Krishnamurthy, Amy Babay.

**Validation:** Konstantinos Pelechrinis.

**Visualization:** Konstantinos Pelechrinis, Xin Liu.

**Writing – original draft:** Konstantinos Pelechrinis, Xin Liu, Prashant Krishnamurthy, Amy Babay.

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
