## [Decision Letter · Decision Letter 0]

6 Feb 2023

PONE-D-22-31512Spotting Anomalous Trades in NFT Markets: The Case of NBA TopshotPLOS ONE

Dear Dr. Pelechrinis,

Thank you for submitting your manuscript to PLOS ONE. After careful consideration, we feel that it has merit but does not fully meet PLOS ONE’s publication criteria as it currently stands. Therefore, we invite you to submit a revised version of the manuscript that addresses the points raised during the review process.

We look forward to receiving your revised manuscript.

Kind regards,

Junhuan Zhang, PhD

Academic Editor

PLOS ONE

Journal Requirements:

Reviewers' comments:

Reviewer's Responses to Questions

**Comments to the Author**

1. Is the manuscript technically sound, and do the data support the conclusions?

Reviewer #1: Partly

Reviewer #2: Yes

2. Has the statistical analysis been performed appropriately and rigorously? 

Reviewer #1: No

Reviewer #2: I Don't Know

3. Have the authors made all data underlying the findings in their manuscript fully available?

Reviewer #1: Yes

Reviewer #2: Yes

4. Is the manuscript presented in an intelligible fashion and written in standard English?

Reviewer #1: Yes

Reviewer #2: Yes

5. Review Comments to the Author

Reviewer #1: The research of NFT is currently in a relatively new and popular situation. This paper studies the NFT market and its transactions from the perspective of abnormal transaction recognition, which is of great significance.

However, in terms of the format and content of the paper, there are still some key points that need to be further improved, so as to support this paper to become a more solid and standardized academic paper.

In terms of the format of the paper, the contents to be improved include the following:

1. In the last paragraph of the first part, the hyperlink references to the content of the following chapters are displayed incorrectly and need to be modified.

2. In addition, the captions and pictures of the paper can be placed together rather than separately. In addition, the clarity of each picture in the paper should also be improved.

3. The content in the subtitles of each section of the paper needs to be enriched with corresponding descriptive text, rather than just one or two sentences. For example, "Circulation Count", "Player", "Bought Price" and so on in Section 2.2.1 need to be enriched.

More importantly, the content of the article needs to be modified, including the following points

1. The data used can be updated. This paper applies the transaction data from 07/27/2020 to 03/19/2021. You can consider applying the latest transaction data to this paper.

2. The application of the conclusion is limited. According to the research conclusion of the paper, it is not possible to accurately evaluate whether a transaction is abnormal.

3. The paper does not consider the user's transaction logic and transaction behavior, but only considers the static transaction price and other information. You can also consider building the behavior model of transaction users/agents.

4. In the result analysis part, the probability threshold (1%) selected in the paper lacks the necessary theoretical support and detailed description.

5. It can be considered further about the underlying blockchain technology corresponding to NFT, such as smart contracts, to enrich the judgment method and conclusion interpretation of abnormal transactions.

There are still quite a few problems in this paper, but the research topic of this paper is of significance. After further revision, I hope this paper can become a good academic paper.

Reviewer #2: The paper is written in a clear and concise manner, with proper technical terms used where necessary and no grammatical errors. The objective and methodology are clearly stated and easy to understand. The topic of NFT markets and their potential for illegal activities is highly relevant and timely, and the focus on the NBA TopShot marketplace is unique and appropriate. The methodology used to build the framework for labeling transactions as anomalous or not is well-explained and appears to be appropriate for the problem at hand.

The results of the network analysis are well presented and provide evidence for the authors' conclusions. However, it would be helpful to have a further explanation of the network metrics used and how they support the findings.

The abstract mentions that the transactions labeled as anomalous will need to be further audited, so it would be helpful to discuss potential avenues for future work.

6. PLOS authors have the option to publish the peer review history of their article (what does this mean?). If published, this will include your full peer review and any attached files.

Reviewer #1: No

Reviewer #2: **Yes: **Wenbin Wu

---

## [Author Response · Author response to Decision Letter 0]

26 Apr 2023

* Response is also provided as a pdf for better formatting and the addition of a figure that cannot be added in the text response **

Reviewer #1 

The research of NFT is currently in a relatively new and popular situation. This paper studies the NFT market and its transactions from the perspective of abnormal transaction recognition, which is of great significance.

However, in terms of the format and content of the paper, there are still some key points that need to be further improved, so as to support this paper to become a more solid and standardized academic paper.

In terms of the format of the paper, the contents to be improved include the following:

1. In the last paragraph of the first part, the hyperlink references to the content of the following chapters are displayed incorrectly and need to be modified.

We have fixed the hype refences in the revised manuscript.

2. In addition, the captions and pictures of the paper can be placed together rather than separately. In addition, the clarity of each picture in the paper should also be improved.

We would like to thank the reviewer for this comment. The separation of the captions and images was according to the journal’s submission guidelines. In fact, we attempted to included them inline with the text in the revised version but the editorial office returned the manuscript with the request to change this. This would have helped to showcase that the quality of the images was simply distorted because they were zoomed in by the submission system during the submission. 

3. The content in the subtitles of each section of the paper needs to be enriched with corresponding descriptive text, rather than just one or two sentences. For example, "Circulation Count", "Player", "Bought Price" and so on in Section 2.2.1 need to be enriched.

We have fixed the subheading for section 2.2.1. However, the mentioned phrases (“Circulation Count”, “Player”, etc.) are not subheadings but rather the names of the variables (explained after the semicolon) used in the model (and presented also in Table 2). 

More importantly, the content of the article needs to be modified, including the following points

1. The data used can be updated. This paper applies the transaction data from 07/27/2020 to 03/19/2021. You can consider applying the latest transaction data to this paper.

We would like to thank the reviewer for this comment. We understand that using more recent data is something that can provide further value but there are a few reasons that we are not able to do so. The practical reason is related to the fact that the platform that provides the transactions (cryptoslam.io) is no longer open/free for use. While we could attempt to obtain the necessary funding to obtain a subscription to their API, the specific platform we are studying (i.e., TopShot), while still active, currently has nowhere near the volume of transactions that our dataset covers. In fact, the transactions declined drastically after 05/2021 (our data ends at 03/19/2021), and the year-to-year decline in the volume has been at 94%. The data we used in the paper includes the vast majority of the lifetime activity of the platform. Even though there is still trading activity on the platform, we believe that our dataset covers a large variety of the different users and transactions that have taken place on the platform. 

2. The application of the conclusion is limited. According to the research conclusion of the paper, it is not possible to accurately evaluate whether a transaction is abnormal.

We completely agree with the reviewer that we cannot evaluate the exact accuracy of the detection. However, this is the case with anomaly detection systems for the majority of the cases, where ground truth is not available. Now for some – well established – tasks/problems (e.g., spam detection) there are benchmark datasets with ground truth that can be used. However, in our case this is clearly not the case. Furthermore, the purpose of “fraud analytics tools” is to build models that will help analysts flag fraudulent activity to be investigated further. This further investigation is usually not automated but rather manual (similar to how IRS audits tax returns – either randomly or after triggers from the system). We have updated the conclusions in the revised manuscript to make this point even more explicit. 

3. The paper does not consider the user's transaction logic and transaction behavior, but only considers the static transaction price and other information. You can also consider building the behavior model of transaction users/agents.

This is certainly a very interesting idea, in that we could create a behavioral model for a user, including “rules” on how they make decision on buying and selling their NFTs. We could then simulate the trading process using this model and look at discrepancies between the actual trade data and the simulations. However, this would again have the limitation of identifying as anomalous the transactions that do not “fit” in the behavioral model we consider. We could possibly learn some of these rules by designing experiments to obtain observations of trades in such a way that will allow us to isolate various behavioral components. However, this is not possiblee and our expertise and strengths do not lay in the design of experiments and behavioral modeling. 

Furthermore, our pricing model can be seen as a behavioral model – to an extent - with respect to the decision on the price a seller is willing to sell, and a buyer is willing to buy a specific NFT. The various model parameters capture the variables that traders are considering when buying/selling. Nevertheless, this model is limited in only including parameters available in our dataset. 

4. In the result analysis part, the probability threshold (1%) selected in the paper lacks the necessary theoretical support and detailed description.

The probability threshold in our framework is very similar to the p-value in traditional hypothesis testing. Specifically, it represents the probability threshold below which we consider the chance of the price having that high/low of a residual as too small to not be anomalous. This probability controls essentially the type I and II errors. Having a higher threshold (e.g., 10%) we will flag more transactions as anomalous and potentially illicit, but a large fraction of them will be “false positives”. With a lower threshold (e.g., 1%) our filtering is more conservative, and we will flag less transactions as possibly illicit with a smaller percentage of them being false positives. However, it is possible that we will have missed flagging more true illicit transactions (i.e., increased “false negatives”). Similarly, with statistical hypothesis testing the choice of threshold is very subjective – and typically, and arbitrarily, selected as 10%, 5%, or 1%. While we have chosen the threshold to be 1%, the most appropriate way for the framework to be used is to output the actual probability of the transaction not fitting a “normal” profile according to the model and let the end user/auditor decide whether this is “small enough” to warrant a flag and further audit.

5. It can be considered further about the underlying blockchain technology corresponding to NFT, such as smart contracts, to enrich the judgment method and conclusion interpretation of abnormal transactions. 

We would like to thank the reviewer for this comment, which indeed can help in detecting abnormal transactions. However, the issue with adopting some of these features in our framework is the fact that our dataset does not include any information related to the smart contract protocols. More importantly, the platform itself – even though it uses a blockchain – is a centralized marketplace, and it is unclear on whether there is a practical use of the various benefits that the underlying technologies can provide. However, in a different system that is really based on a distributed ledger with smart contracts, features obtained by the various protocol variables can be extremely helpful, possibly in the auditing of the anomalous transactions identified by the “first level” of anomaly detection. We have provided additional details on this in the revised manuscript. 

There are still quite a few problems in this paper, but the research topic of this paper is of significance. After further revision, I hope this paper can become a good academic paper.

Reviewer #2 

The paper is written in a clear and concise manner, with proper technical terms used where necessary and no grammatical errors. The objective and methodology are clearly stated and easy to understand. The topic of NFT markets and their potential for illegal activities is highly relevant and timely, and the focus on the NBA TopShot marketplace is unique and appropriate. The methodology used to build the framework for labeling transactions as anomalous or not is well-explained and appears to be appropriate for the problem at hand. 

1. The results of the network analysis are well presented and provide evidence for the authors' conclusions. However, it would be helpful to have a further explanation of the network metrics used and how they support the findings.

We would like to thank the reviewer for this comment. We have added some details and information about the various metrics that we used that we believe will help the readers that are not familiar with network science understand them better.

2. The abstract mentions that the transactions labeled as anomalous will need to be further audited, so it would be helpful to discuss potential avenues for future work.

We would like to thank the reviewer for this comment, and accordingly we have updated the conclusions in the revised manuscript to provide some more on what this further auditing might entail, which can also serve as a roadmap for future work.

---

## [Decision Letter · Decision Letter 1]

2 Jun 2023

Spotting Anomalous Trades in NFT Markets: The Case of NBA Topshot

PONE-D-22-31512R1

Dear Dr. Pelechrinis,

We’re pleased to inform you that your manuscript has been judged scientifically suitable for publication and will be formally accepted for publication once it meets all outstanding technical requirements.

Kind regards,

Junhuan Zhang, PhD

Academic Editor

PLOS ONE

Additional Editor Comments (optional):

Reviewers' comments:

Reviewer's Responses to Questions

**Comments to the Author**

1. If the authors have adequately addressed your comments raised in a previous round of review and you feel that this manuscript is now acceptable for publication, you may indicate that here to bypass the “Comments to the Author” section, enter your conflict of interest statement in the “Confidential to Editor” section, and submit your "Accept" recommendation.

Reviewer #1: (No Response)

Reviewer #2: All comments have been addressed

2. Is the manuscript technically sound, and do the data support the conclusions?

Reviewer #1: Yes

Reviewer #2: Yes

3. Has the statistical analysis been performed appropriately and rigorously? 

Reviewer #1: Yes

Reviewer #2: Yes

4. Have the authors made all data underlying the findings in their manuscript fully available?

Reviewer #1: Yes

Reviewer #2: Yes

5. Is the manuscript presented in an intelligible fashion and written in standard English?

Reviewer #1: Yes

Reviewer #2: Yes

6. Review Comments to the Author

Reviewer #1: (No Response)

Reviewer #2: (No Response)

7. PLOS authors have the option to publish the peer review history of their article (what does this mean?). If published, this will include your full peer review and any attached files.

Reviewer #1: No

Reviewer #2: **Yes: **Wenbin Wu

---

## [Editor Report · Acceptance letter]

6 Jun 2023

PONE-D-22-31512R1 

Spotting Anomalous Trades in NFT Markets: The Case of
NBA Topshot 

Dear Dr. Pelechrinis:

I'm pleased to inform you that your manuscript has been deemed suitable for publication in PLOS ONE. Congratulations! Your manuscript is now with our production department. 

Kind regards, 

on behalf of

Dr. Junhuan Zhang 

Academic Editor

PLOS ONE